# From Broad Exploration to Stable Synthesis: Entropy-Guided Optimization for Autoregressive Image Generation

**Han Song**[1*], **Yucheng Zhou**[2*], **Jianbing Shen**[2], **Yu Cheng**[1✉]
[1]The Chinese University of Hong Kong, [2]University of Macau
`hansong@link.cuhk.edu.hk, yucheng.zhou@connect.um.edu.mo`
`chengyu@cse.cuhk.edu.hk`

## Abstract

Combining Chain-of-Thought (CoT) with Reinforcement Learning (RL) improves text-to-image (T2I) generation, yet the underlying interaction between CoT's exploration and RL's optimization remains unclear. We present a systematic entropy-based analysis that yields three key insights: (1) CoT expands the generative exploration space, while RL contracts it toward high-reward regions; (2) final reward is strongly negatively correlated with both the mean and variance of image-token entropy, highlighting the need to reduce uncertainty and instability; and (3) the entropy of the textual CoT directly governs downstream image quality, with lower-entropy CoTs leading to better generations. Motivated by these findings, we propose *Entropy-Guided Group Relative Policy Optimization* (EG-GRPO), a fine-tuning strategy that reallocates optimization budget by uncertainty: low-entropy tokens are excluded from reward-driven updates to preserve stability, while high-entropy tokens receive an entropy bonus that encourages structured exploration without collapse. Experiments on standard T2I benchmarks demonstrate that EG-GRPO achieves state-of-the-art performance. Our code is available at `https://github.com/minebetter/EG-GRPO`.

## 1 Introduction

Text-to-image (T2I) generation has progressed rapidly with large-scale pretraining and strong autoregressive and diffusion architectures (Chen et al., 2023; Labs, 2024; Wang et al., 2024; Zhou et al., 2024), yet two core challenges remain: (i) balancing *exploration* for diversity against *exploitation* for reward-aligned fidelity, and (ii) ensuring *generation stability* under repeated sampling. Chain-of-Thought (CoT) prompting promises richer semantic planning (Wei et al., 2022), while reinforcement learning (RL) directly optimizes task or preference rewards (Jiang et al., 2025). However, how CoT's exploratory behavior interacts with RL's optimization in T2I, and how this interaction governs uncertainty and stability, has not been systematically understood.

We analyze *entropy dynamics* in autoregressive T2I models that combine CoT with RL (via Group Relative Policy Optimization, GRPO (Jiang et al., 2025)) and use Shannon entropy to quantify token-level uncertainty in both modalities: textual CoT tokens and image tokens. Three empirical findings emerge. First, CoT *expands* the exploration space, broadening the entropy distribution of generated outputs, whereas RL *contracts* this space toward higher-reward regions. Second, final reward exhibits a strong negative correlation with both the *mean* and the *standard deviation* of image-token entropy, indicating that reducing uncertainty and instability is central to quality. Third, the entropy of the textual CoT *directly* influences downstream image quality: lower-entropy CoTs yield tighter, higher-reward clusters under stable sampling.

Guided by these findings, we propose *Entropy-Guided Group Relative Policy Optimization (EG-GRPO)*, a token-level modification of GRPO that reallocates gradient budget by uncertainty. Low-entropy (high-confidence) tokens are excluded from reward-driven updates, retaining only the KL-

---

*Equal contribution.
✉Corresponding Author.

to-reference term to preserve stability and previously acquired knowledge. High-entropy tokens receive an *entropy bonus* added to the advantage, encouraging structured exploration and accelerating uncertainty reduction where it matters. A batch-level calibration ties the bonus magnitude to the mass saved on low-entropy tokens, keeping update scale close to GRPO, and the bonus vanishes at GRPO equilibrium, preserving the stationary points of the base objective (Wang et al., 2025c).

We evaluate EG-GRPO on T2I-CompBench (Huang et al., 2023) and WISE (Niu et al., 2025) using a Janus-Pro autoregressive backbone in a discrete latent space (Chen et al., 2025) and a standard reward pipeline combining human-preference scoring, object grounding, and VQA signals (Wu et al., 2023; Liu et al., 2024; Wang et al., 2022). EG-GRPO attains state-of-the-art results, with pronounced gains in compositional generalization (e.g., attribute binding and object relations). Ablations that apply entropy guidance to only CoT tokens or only image tokens underperform the full model, confirming the need to control uncertainty in both semantic planning and visual decoding.

The main contributions of our paper can be summarized as follows:

- We provide a quantitative account of the CoT–RL interaction through entropy dynamics: CoT expands exploration, RL contracts toward high-reward regions; reward is strongly negatively correlated with entropy mean and std of ima; and textual CoT entropy governs downstream image quality.

- We introduce an entropy-guided, token-level refinement of GRPO that protects confident tokens via KL-only updates and focuses optimization on uncertain tokens via an entropy bonus, with calibrated budget and equilibrium-vanishing properties.

- On T2I-CompBench and WISE, EG-GRPO achieves state-of-the-art performance and reduces uncertainty and instability in line with the analysis.

## 2 RELATED WORK

### 2.1 TEXT-TO-IMAGE GENERATION MODEL

Text-to-image generation has been advanced along two major paradigms: autoregressive modeling and diffusion-based approaches (Team et al., 2025; Zhou et al., 2025). On the autoregressive side, Parti (Yu et al., 2022) demonstrates that large-scale transformer models can achieve impressive compositional generation by predicting image tokens sequentially. Fluid (Fan et al., 2024) further explores continuous tokens and generation order, showing improved efficiency and quality. STAR (Ma et al., 2024) introduces a scale-wise autoregressive framework that progressively generates images from coarse to fine scales, while JetFormer (Tschannen et al., 2024) directly models raw images and text in a unified autoregressive manner without discrete tokenization. More recently, NextStep-1 (Team et al., 2025) scales continuous-token autoregressive generation to 14B parameters, achieving strong performance in both image synthesis and editing. In parallel, diffusion models have become dominant for high-quality synthesis. VQ-Diffusion (Gu et al., 2022) combines vector-quantized representations with diffusion for discrete latent modeling, and ERNIE-ViLG 2.0 (Feng et al., 2023) extends this to large-scale multilingual text-to-image generation. UPainting (Li et al., 2022) introduces cross-modal guidance to unify simple and complex scenarios, while RPG (Yang et al., 2024) enhances controllability via recaptioning, planning, and region-based diffusion. These works illustrate the complementary strengths of autoregressive and diffusion frameworks in advancing the controllability, fidelity, and scalability of text-to-image generation.

### 2.2 CHAIN OF THINKING AND REINFORCEMENT LEARNING

Recent works have explored integrating Chain-of-Thought (CoT) reasoning with reinforcement learning (RL) to improve text-to-image generation. Visual-CoG (Li et al., 2025) introduces stage-aware RL with intermediate rewards across semantic planning, refinement, and evaluation, while ReasonGen-R1 (Zhang et al., 2025b) and T2I-R1 (Jiang et al., 2025) incorporate rationale-augmented data and bi-level reasoning chains optimized via GRPO. PromptEnhancer (Wang et al., 2025b) further demonstrates that CoT-based prompt rewriting with RL can enhance image quality without modifying the generator, and verification-based methods also inject preference alignment during synthesis (Zhang et al., 2025a). Beyond explicit CoT, RL has been applied for alignment

in diffusion models, such as comparing DPO and GRPO (Tong et al., 2025), subject-driven preference optimization (Miao et al., 2024), and DPOK fine-tuning with KL regularization (Fan et al., 2023). Related multimodal reasoning approaches, including ImageGen-CoT (Liao et al., 2025) and reflective CoT for retrieval (Wu et al., 2024), highlight the broader potential of structured reasoning. Combining CoT and RL provides a promising avenue for enhancing controllability, interpretability, and human alignment in text-to-image generation. While SimpleAR (Wang et al., 2025a) and Gallici & Borde (2025) demonstrate the efficacy of standard GRPO for high-fidelity and style-aligned autoregressive generation, our approach introduces an entropy-guided mechanism to reallocate the optimization budget at the token level dynamically.

## 3 PRELIMINARIES

We begin by formalizing autoregressive text-to-image generation, introducing Shannon entropy as a measure of generative uncertainty, and reviewing Group Relative Policy Optimization (GRPO) for model refinement.

### 3.1 AUTOREGRESSIVE GENERATION IN DISCRETE LATENT SPACE

Autoregressive text-to-image models operate in a discrete latent space. An image $I$ is first encoded into tokens $z = (z_1, z_2, \ldots, z_L)$ via a pre-trained tokenizer such as VQ-VAE. The conditional likelihood given a text prompt $c$ factorizes autoregressively:

$$p(z \mid c) = \prod_{i=1}^{L} p(z_i \mid z_{<i}, c). \tag{1}$$

A policy $\pi_\theta$ parameterized by $\theta$ models these conditionals. At step $i$, the policy outputs a categorical distribution $\pi_\theta(\cdot \mid z_{<i}, c)$ over the vocabulary $\mathcal{V}$, from which $z_i$ is sampled. The final image is reconstructed by decoding the full sequence $z$.

### 3.2 ENTROPY AS PREDICTIVE UNCERTAINTY

For each step $i$, the policy distribution admits a Shannon entropy:

$$H(\pi_\theta(\cdot \mid z_{<i}, c)) = -\sum_{j \in \mathcal{V}} p_j \log p_j, \tag{2}$$

where $p_j$ denotes the probability of token $j$. Low entropy reflects confident, often high-fidelity predictions with reduced diversity, whereas high entropy reflects uncertainty, encouraging exploration and diverse generations at the cost of possible incoherence.

### 3.3 GROUP RELATIVE POLICY OPTIMIZATION

Group Relative Policy Optimization (GRPO) refines generative policies using relative rewards without requiring a value function. For a prompt $c$, the policy $\pi_\theta$ samples $G$ candidate sequences

$$\{o^{(i)}\}_{i=1}^{G} \sim \pi_\theta(\cdot \mid c), \quad o^{(i)} = (o_1^{(i)}, \ldots, o_{T^{(i)}}^{(i)}). \tag{3}$$

Each sequence receives a reward $r^{(i)}$, which is normalized within the group:

$$\mu = \tfrac{1}{G} \sum_{i=1}^{G} r^{(i)}, \quad \sigma = \sqrt{\tfrac{1}{G} \sum_{i=1}^{G} (r^{(i)} - \mu)^2}, \quad A^{(i)} = \tfrac{r^{(i)} - \mu}{\max\{\sigma, \varepsilon\}}. \tag{4}$$

Broadcasting $A^{(i)}$ to all tokens yields the training objective

$$\mathcal{L}_{\text{GRPO}}(\theta) = -\tfrac{1}{G} \sum_{i=1}^{G} \tfrac{1}{T^{(i)}} \sum_{t=1}^{T^{(i)}} A^{(i)} \log \pi_\theta(o_t^{(i)} \mid c, o_{<t}^{(i)}) + \beta \, D_{\text{KL}}(\pi_\theta \,\|\, \pi_{\text{ref}}), \tag{5}$$

where $\beta \geq 0$ controls a KL regularizer toward a reference policy. This group-based normalization yields scale-invariant advantages and stable updates, making GRPO well-suited for aligning autoregressive generators with task-specific rewards.

# 4 ANALYSIS OF ENTROPY DYNAMICS IN TEXT-TO-IMAGE GENERATION

To investigate the role of uncertainty in text-to-image generation, we analyze *generative entropy* as a quantitative indicator of model behavior. Our study focuses on disentangling how CoT and RL fine-tuning affect entropy within the pipeline. By examining their individual and combined effects, we clarify how these techniques balance exploration and exploitation, and how the resulting entropy dynamics define the optimization objective for high-quality visual synthesis.

## 4.1 THE DICHOTOMY OF EXPLORATION AND EXPLOITATION: CoT VS. RL

We begin our analysis by examining the distinct yet complementary roles of Chain-of-Thought (CoT) prompting and reinforcement learning (RL) fine-tuning. For each textual prompt, we generate multiple image candidates under three settings: the baseline model (*Janus-Pro*), the baseline augmented with CoT reasoning (*Janus-Pro+CoT*), and a GRPO-finetuned variant built upon *Janus-Pro+CoT* (*T2I-R1*). To investigate their differences, we assess each generated image by jointly measuring its output mean entropy and reward score, and visualize the resulting distributions in a two-dimensional space.

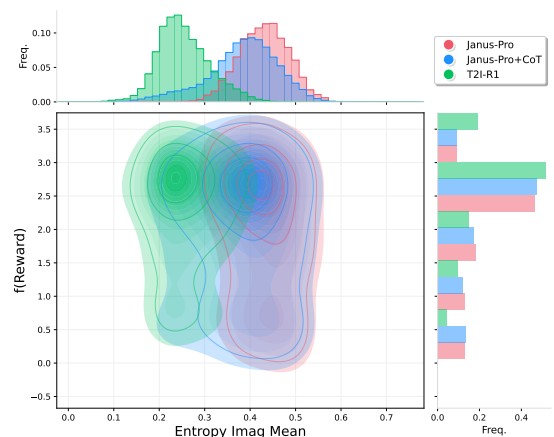

Figure 1: Comparison of different text-to-image generation methods: (a) autoregressive text-to-image generation, (b) CoT, and (c) with CoT and GRPO optimization.

As illustrated in Figure 2, the introduction of CoT significantly broadens the entropy distribution of the generated images. It shows that CoT expands the model's exploratory space, enabling it to generate a more diverse set of outputs. This expanded space contains both low- and high-reward samples, indicating that CoT itself does not guarantee quality but rather increases the range of generative possibilities. Conversely, the application of Group Relative Policy Optimization (GRPO), which results in the T2I-R1, leads to a notable contraction and leftward shift of the entropy distribution. This demonstrates that the model learns to exploit the vast space unlocked by CoT, converging towards a much narrower, lower-entropy region that consistently yields higher rewards. This reveals a complementary relationship: CoT serves to **expand the exploratory landscape**, while RL acts as a refinement mechanism to **exploit this landscape and guide the model towards stable, high-quality regions**.

Figure 2: Entropy–reward distributions of different methods. CoT (*Janus-Pro+CoT*) expands the exploratory space with more diverse outputs, while GRPO fine-tuning (*T2I-R1*) contracts it toward higher-reward regions, yielding more stabilized, high-quality generations.

## 4.2 UPSTREAM INFLUENCE: HOW CoT'S TEXTUAL ENTROPY GOVERNS IMAGE QUALITY

Since Chain-of-Thought serves as the entry point of exploration, we further examine how the intrinsic uncertainty of the textual CoT affects downstream image generation. To isolate this effect, we focus on a subset of samples where image generation remains stable, defined as those with entropy variance below a small threshold of 0.011 across multiple runs. This filtering minimizes confounding factors due to unstable sampling, allowing us to probe the impact of CoT entropy directly.

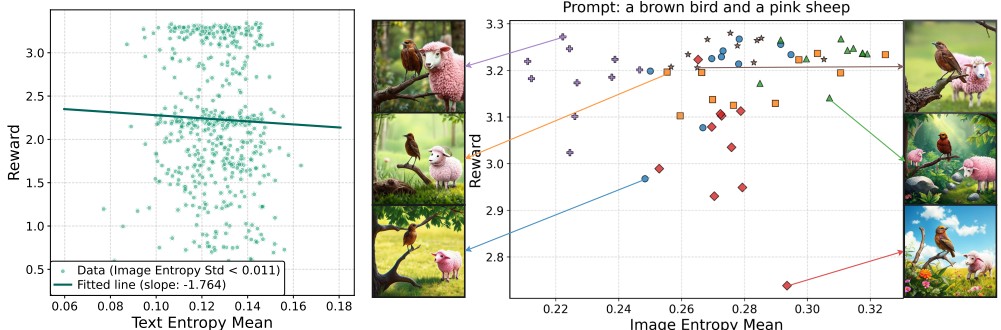

Figure 3: **Left:** Reward vs. CoT entropy (stable cases, Image Entropy Std < 0.011). Higher CoT entropy correlates with lower image reward. **Right:** Reward distributions across different CoTs for the same prompt. Images from the same CoT cluster together, with certain CoTs consistently yielding lower rewards.

As shown in Figure 3 (Left), we observe a clear negative correlation between the mean entropy of the textual CoT and the average reward of the resulting images. Intuitively, high-entropy CoTs correspond to less coherent or more uncertain reasoning traces, which tend to degrade visual quality. In contrast, low-entropy CoTs provide more confident and consistent reasoning, ultimately leading to higher-reward generations. This analysis is grounded on the stable subset, ensuring that the observed trend is not an artifact of sampling instability.

To further dissect this phenomenon, we visualize the reward distributions of images conditioned on different CoTs for the same prompt (Figure 3, Right). Each distinct CoT forms a compact cluster in the reward space, highlighting its consistent influence on generation outcomes. Notably, certain CoTs repeatedly produce clusters with lower average rewards, suggesting that the quality bottleneck is determined upstream, at the textual reasoning stage. These findings demonstrate a direct transmission of uncertainty from text to image: the entropy of CoT reasoning acts as a critical upstream factor that governs the attainable quality of visual outputs.

## 4.3 THE LEARNED OBJECTIVE: MINIMIZING UNCERTAINTY AND INSTABILITY FOR HIGHER REWARDS

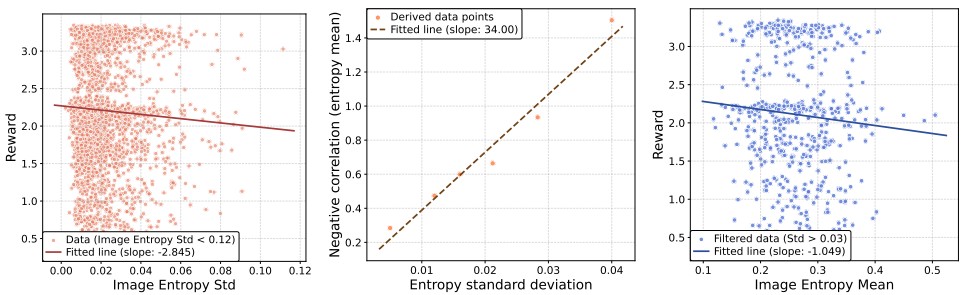

Figure 4: **Left:** Reward vs. entropy std. Higher instability (larger std) consistently lowers reward. **Middle:** Relation between entropy std (x-axis) and the negative correlation of reward–entropy mean (y-axis). Greater instability strengthens the negative correlation. **Right:** Reward vs. entropy mean under high-variance cases (std > 0.03). Large std implies exploratory generation where RL has not converged; in this regime, reducing mean entropy is especially beneficial.

We next investigate how the fully RL-trained model internalizes entropy control to optimize for reward. For each prompt–CoT pair, multiple images are generated, and their entropy statistics are aggregated. Specifically, the mean entropy characterizes the global level of uncertainty in the generative process, while the standard deviation (std) captures instability across different runs.

Empirical results reveal a consistent negative correlation between reward and standard deviation of entropy (Figure 4, Left). This indicates that generative instability is inherently detrimental: mod-

els that produce more consistent entropy trajectories across samples yield higher-quality outputs. Beyond this first-order observation, we further examine how instability modulates the role of uncertainty. Figure 4 (Middle) demonstrates that the negative correlation between mean entropy and reward becomes progressively stronger as std increases. In other words, instability amplifies the adverse impact of uncertainty, rendering mean entropy a more decisive factor under high-variance conditions.

To isolate this effect, we analyze the high-variance regime (std $> 0.03$), where the image token generation remains exploratory and the RL policy has not yet converged. As shown in Figure 4 (Right), in such cases, lowering mean entropy proves particularly effective in improving reward. This finding suggests that, when the model operates in an unstable exploratory state, suppressing overall uncertainty is a critical pathway toward higher-quality generations.

These analyses indicate that the RL agent implicitly optimizes a compound objective: **simultaneously minimizing overall uncertainty (mean entropy) and reducing instability across runs (entropy std)**.

## 5 ENTROPY-GUIDED GROUP RELATIVE POLICY OPTIMIZATION

Building on Section 4, we introduce a token-level optimization scheme that preserves GRPO's group-relative structure while reallocating updates toward uncertain parts of the generation process. *Entropy-Guided GRPO (EG-GRPO)* applies to both textual CoT and image tokens and adds an entropy-based bonus only where uncertainty is high.

### 5.1 DESIGN PRINCIPLES FROM ENTROPY ANALYSIS

Section 4 showed that: (i) CoT broadens exploration while RL contracts it toward high-reward regions; (ii) rewards are negatively correlated with both the mean and the variance of token entropies; and (iii) textual (CoT) entropy causally influences downstream image quality. We adopt three principles:

1. **Focus on uncertainty.** Allocate more update mass to high-entropy tokens to reduce instability where it matters.
2. **Protect confidence.** On the lowest-entropy tokens, set the reward-driven advantage to zero so that only the KL-to-reference acts, preventing drift on confident regions.
3. **Stay reward-driven.** Retain the GRPO group-relative objective; entropy contributes an *additive* per-token bonus at high entropy without replacing advantage.

### 5.2 REVISITING GRPO AND TOKEN BROADCAST

For prompt $c$, policy $\pi_\theta$ samples $G$ sequences $\{o^{(i)}\}_{i=1}^G$ with rewards $\{r^{(i)}\}$ normalized to group-relative advantages $A^{(i)}$. GRPO optimizes

$$\mathcal{L}_{\text{GRPO}}(\theta) = -\frac{1}{G}\sum_{i=1}^G \frac{1}{T^{(i)}} \sum_{t=1}^{T^{(i)}} A^{(i)} \log \pi_\theta\big(o_t^{(i)} \mid c, o_{<t}^{(i)}\big) \; + \; \beta\, D_{\text{KL}}\big(\pi_\theta \,\|\, \pi_{\text{ref}}\big). \tag{6}$$

This *broadcasts* the same coefficient $A^{(i)}$ to all tokens of sequence $i$, ignoring per-token uncertainty and potentially wasting gradient budget on already-confident tokens.

### 5.3 ENTROPY-GUIDED TOKEN SELECTION

Let $H_t^{(i)} \triangleq H\big(\pi_\theta(\cdot \mid c, o_{<t}^{(i)})\big)$ be the Shannon entropy at token $t$ of sequence $i$, and define the normalized entropy $\bar{H}_t^{(i)} \triangleq H_t^{(i)} / \log|\mathcal{V}| \in [0,1]$. For each sequence $i$ and each modality $m \in \{\text{text}, \text{image}\}$ independently,[1] compute per-sequence percentiles and define

$$\mathcal{S}_{\text{hi}}^{(i,m)} = \text{top-50\% by } \bar{H}_t^{(i)}, \quad \mathcal{S}_{\text{lo}}^{(i,m)} = \text{bottom-20\%}, \quad \mathcal{S}_{\text{mid}}^{(i,m)} = \text{remaining}.$$

---

[1] We rank and threshold entropies on textual CoT and image tokens separately, consistent with Section 4.

Introduce masks $M_t^{(i)}, U_t^{(i)} \in \{0, 1\}$:

$$M_t^{(i)} = \mathbb{I}[t \notin \mathcal{S}_{\text{lo}}^{(i,m)}], \qquad U_t^{(i)} = \mathbb{I}[t \in \mathcal{S}_{\text{hi}}^{(i,m)}],$$

so $M_t^{(i)} = 0$ removes reward-driven updates on low-entropy tokens, and $U_t^{(i)} = 1$ marks high-entropy tokens to receive a bonus.

**Budget view.** Let $p_{\text{lo}}, p_{\text{mid}}, p_{\text{hi}}$ be the fractions of tokens in $\mathcal{S}_{\text{lo}}^{(i,m)}, \mathcal{S}_{\text{mid}}^{(i,m)}, \mathcal{S}_{\text{hi}}^{(i,m)}$ ($p_{\text{hi}} = 0.5$, $p_{\text{lo}} = 0.2$, $p_{\text{mid}} = 0.3$). GRPO's per-sequence coefficient budget is $B_{\text{GRPO}}^{(i)} \triangleq \frac{1}{T^{(i)}} \sum_{t=1}^{T^{(i)}} |A^{(i)}| = |A^{(i)}|$. Under EG-GRPO,

$$B_{\text{EG}}^{(i)} \triangleq \frac{1}{T^{(i)}} \sum_{t=1}^{T^{(i)}} \big| M_t^{(i)} A^{(i)} + U_t^{(i)} \lambda \, \text{sg}[\bar{H}_t^{(i)}] \big|, \tag{7}$$

with $\lambda \geq 0$ and stop-gradient $\text{sg}[\cdot]$. Zeroing low-entropy updates *saves* roughly $p_{\text{lo}}|A^{(i)}|$ of mass and *reinvests* it on high-entropy tokens through the additive bonus $\lambda \, \text{sg}[\bar{H}_t^{(i)}]$.

**Proposition 1** (Per-batch budget balance)**.** *For a batch $\mathcal{B}$, choose*

$$\lambda^\star \triangleq \kappa \cdot \frac{\sum_{i \in \mathcal{B}} |A^{(i)}| \cdot \frac{1}{T^{(i)}} \sum_{t \in \mathcal{S}_{\text{lo}}^{(i,m)}} 1}{\sum_{i \in \mathcal{B}} \frac{1}{T^{(i)}} \sum_{t \in \mathcal{S}_{\text{hi}}^{(i,m)}} \text{sg}[\bar{H}_t^{(i)}]} \quad \text{with } \kappa \in (0, 1]. \tag{8}$$

*Then $\mathbb{E}_{\mathcal{B}}[B_{\text{EG}}^{(i)}] \approx \kappa \cdot \mathbb{E}_{\mathcal{B}}[B_{\text{GRPO}}^{(i)}]$. A detailed derivation and calibration discussion are deferred to Appendix B.1. Setting $\kappa = 1$ yields batch-level budget neutrality in the calibrated upper-bound sense.*

**Fixed-point neutrality.** Because $\lambda^\star$ scales with $\sum_i |A^{(i)}|$, the bonus vanishes at GRPO equilibrium where group-relative advantages cancel. See Appendix B.2 for a formal proof.

**Corollary 5.1** (Preserving GRPO stationary points)**.** *If $A^{(i)} \equiv 0$ for all $i$ and $\lambda = \lambda^\star$ with $\kappa = 1$, then $\lambda^\star = 0$ and $\tilde{A}_t^{(i)} \equiv 0$ for all $t$; EG-GRPO reduces to the KL regularizer and preserves the stationary point.*

## 5.4 ENTROPY-BIASED ADVANTAGE

We modify the broadcasted coefficient at token $t$ of sequence $i$ by

$$\tilde{A}_t^{(i)} = M_t^{(i)} A^{(i)} + U_t^{(i)} \lambda \, \text{sg}[\bar{H}_t^{(i)}], \tag{9}$$

where $M_t^{(i)}$ removes reward-driven updates on the lowest-entropy 20% tokens and $U_t^{(i)}$ adds an entropy bonus on the highest-entropy 50% tokens. The EG-GRPO loss is

$$\mathcal{L}_{\text{EG-GRPO}}(\theta) = -\frac{1}{G} \sum_{i=1}^{G} \frac{1}{T^{(i)}} \sum_{t=1}^{T^{(i)}} \tilde{A}_t^{(i)} \log \pi_\theta\big(o_t^{(i)} \mid c, o_{<t}^{(i)}\big) + \beta \, D_{\text{KL}}\big(\pi_\theta \,\|\, \pi_{\text{ref}}\big). \tag{10}$$

$\beta$ and the reference-policy KL are unchanged; low-entropy tokens are therefore governed solely by KL when $M_t^{(i)} = 0$.

**Reward-shaping view.** Define a token-level pseudo-reward $\tilde{r}_t^{(i)} \triangleq r_{\text{grp}}^{(i)} \cdot M_t^{(i)} + \lambda \, \text{sg}[\bar{H}_t^{(i)}] \cdot U_t^{(i)}$, where $r_{\text{grp}}^{(i)}$ induces $A^{(i)}$. Then $\mathbb{E}\big[\nabla_\theta \log \pi_\theta(o_t^{(i)} \mid \cdot) \tilde{r}_t^{(i)}\big]$ is an unbiased policy-gradient estimator for an augmented objective whose baseline component is GRPO.

## 5.5 WHY EG-GRPO REDUCES UNCERTAINTY AND PRESERVES KNOWLEDGE

**(A) Targeted entropy reduction.** For small steps, the update at token $t$ is proportional to $\tilde{A}_t^{(i)} \nabla_\theta \log \pi_\theta(o_t^{(i)} \mid \cdot)$. On high-entropy tokens, $\tilde{A}_t^{(i)} = A^{(i)} + \lambda \, \text{sg}[\bar{H}_t^{(i)}]$ strengthens positive updates and attenuates negative ones, lowering entropy where it is largest under softmax parameterizations.

**(B) Stability on confident tokens.** When $M_t^{(i)} = 0$, reward-driven gradients vanish and only KL-to-reference remains, protecting confident regions from drift and preserving learned knowledge.

**(C) Proximity to GRPO equilibrium.** With $\lambda = \lambda^\star$ and $\kappa = 1$, the batch-wise coefficient budget matches GRPO (Appendix B.1), and the bonus disappears at equilibrium (Appendix B.2), reallocating update mass without altering stationary points.

## 6 EXPERIMENTS

### 6.1 EXPERIMENTAL SETTINGS

**Training Details.** Following (Jiang et al., 2025), we train our policy on the same 6,786 text prompts drawn from T2I-CompBench (Huang et al., 2023), which contain only textual descriptions without paired images. The prompts are accompanied by structured object–attribute annotations that were automatically extracted using GPT-4o mini in prior work (Jiang et al., 2025). The policy backbone is initialized from Janus-Pro-7B (Chen et al., 2025). Optimization uses a learning rate of $1 \times 10^{-6}$ and a KL coefficient $\beta = 0.01$. For the reward pipeline, we combine HPS (Wu et al., 2023) as the human-preference estimator, GroundingDINO (Liu et al., 2024) as the object detector, and GIT (Wang et al., 2022) as the VQA model. The object–relation module (ORM) is implemented by finetuning LLaVA-OneVision-7B following the procedure of Guo et al. (2025).

**Benchmark.** To assess the effectiveness of our approach, we rely on two established evaluation suites: T2I-CompBench (Huang et al., 2023) and WISE (Niu et al., 2025). T2I-CompBench provides 6,000 prompts designed to test compositional generalization. The benchmark covers three broad categories: attribute binding, object relations, and complex compositions, which are further split into six sub-categories, such as color/shape/texture bindings, spatial and non-spatial relations, and multi-object compositions. In contrast, WISE focuses on knowledge-intensive reasoning. It contains 1,000 prompts spanning cultural commonsense, spatial–temporal reasoning, and natural science, requiring the model to infer what specific entity or situation should appear in the image. For WISE, since the corresponding reasoning instructions from prior work (Jiang et al., 2025) were not released, we reimplemented them ourselves and provide the exact templates in the Appendix C for transparency. For both benchmarks, we otherwise follow the official evaluation protocols.

### 6.2 MAIN RESULTS

| Model | T2I-CompBench | | | WISE | | |
|---|---|---|---|---|---|---|
| | **Color** | **Shape** | **Texture** | **Culture** | **Spatio-temporal** | **Science** |
| PixArt-$\alpha$ (Chen et al., 2023) | 66.90 | 49.27 | 64.77 | 45.00 | 49.00 | **46.33** |
| SD-v1.5 (Rombach et al., 2022) | 37.58 | 37.13 | 41.86 | 34.00 | 33.50 | 26.00 |
| FLUX.1-dev (Labs, 2024) | 74.07 | 57.18 | 69.22 | 48.00 | **60.00** | 42.67 |
| Emu3 (Wang et al., 2024) | 75.44 | 57.06 | 71.64 | 34.00 | 46.50 | 37.67 |
| Show-o (Xie et al., 2024) | 56.00 | 41.00 | 46.00 | 28.00 | 44.00 | 35.33 |
| Janus-Pro-7B (Chen et al., 2025) | 63.59 | 35.28 | 49.36 | 30.00 | 43.00 | 34.67 |
| T2I-R1* (Jiang et al., 2025) | 82.58 | 58.67 | 76.94 | 48.00 | 55.50 | 45.00 |
| EG-GRPO (Ours) | **84.11** | **60.88** | **77.38** | **49.00** | 56.00 | **46.33** |

Table 1: Comparison of models across **T2I-CompBench** and **WISE**. Spatio-temporal is the average of Time and Space; Science is the average of Biology, Physics, and Chemistry. *Results are obtained by evaluating the officially released model under the same experimental settings for fair comparison.

As shown in Table 1, EG-GRPO achieves the strongest results on T2I-CompBench, surpassing all baselines in Color (84.11), Shape (60.88), and Texture (77.38), with particularly notable gains on Shape binding. On WISE, EG-GRPO improves over T2I-R1 in Culture (49.00 vs. 48.00) and Science (46.33 vs. 45.00) while maintaining comparable performance in Spatio-temporal (56.00 vs. 55.50). These consistent improvements demonstrate that entropy-guided updates effectively enhance compositional reasoning and robustness while preserving the stability of knowledge learned by the base model.

| Model | T2I-CompBench | | | WISE | | |
|---|---|---|---|---|---|---|
| | Color | Shape | Texture | Culture | Spatio-temporal | Science |
| EG-GRPO | 84.11 | 60.88 | 77.38 | 49.00 | 56.00 | 46.33 |
| w/ only sem | 81.29 | 55.68 | 74.10 | 47.00 | 56.00 | 43.67 |
| w/ only tok | 79.25 | 53.73 | 72.46 | 45.00 | 56.00 | 43.00 |
| w/o All | 82.58 | 58.67 | 76.94 | 48.00 | 55.50 | 45.00 |

Table 2: Ablation results of EG-GRPO. We compare the full method with variants that apply entropy guidance only to textual CoT tokens (*w/ only sem*), only to image tokens (*w/ only tok*), or not at all (*w/o All*).

## 6.3 ABLATION STUDY

Table 2 summarizes the effect of applying entropy guidance on different token types. The full EG-GRPO model, which adds entropy bonuses to both textual CoT tokens and image tokens, achieves the best overall performance. In contrast, applying entropy guidance to only textual CoT tokens (*w/ only sem*) or only image tokens (*w/ only tok*) leads to degraded performance. This suggests that introducing entropy regulation in a single modality can create an imbalance during optimization, which may be more harmful than applying no entropy guidance at all. The baseline without entropy guidance (*w/o All*) performs worse than the full EG-GRPO model, confirming that the proposed entropy-aware updates are essential for improved compositional generalization.

## 6.4 ANALYSIS ON DIVERSITY

In this section, we address the trade-off between entropy reduction and generative diversity. While entropy is a measure of uncertainty, reducing it raises the question of whether the model's creative expressivity and output diversity are negatively impacted. To investigate this, we employ the **Vendi Score** (Friedman & Dieng, 2023), a reference-free metric designed to quantify diversity in machine learning models. Additional qualitative diversity comparisons are provided in Sec. E.

**Dynamics of Diversity during RL.** The RL naturally contracts the exploration space to focus on high-reward regions. As shown in Table 3, our analysis of the GRPO training dynamics confirms this expected behavior: as the model optimizes for reward-aligned fidelity from Step 100 to 800, the diversity naturally shows a slight decrease.

| Step | Diversity (Vendi Score) |
|---|---|
| 100 | 2.7305 |
| 200 | 2.7233 |
| 400 | 2.7212 |
| 600 | 2.7151 |
| 800 | 2.7159 |

Table 3: Evolution of diversity during the GRPO training process. As the model converges towards high-reward regions, a slight reduction in diversity is observed, reflecting the exploration-exploitation trade-off.

**Preserving Diversity at Similar Quality.** Crucially, our method targets *instability* rather than *semantic diversity*. To verify this, we compared EG-GRPO against the baseline (T2I-R1) on a filtered subset of generated samples where both models achieved similar quality scores (defined as $|\Delta \text{Quality}| < 0.1$). The quality score is an aggregate of BLIP-2 (Li et al., 2023), LAION-Aesthetics (Beaumont et al., 2022), and PickScore (Kirstain et al., 2023).

As shown in Table 4, EG-GRPO maintains a Vendi Score (2.593) virtually identical to the baseline (2.592) under the same quality constraints. This indicates that the Entropy Bonus (Section 5.3) successfully preserves valid exploration pathways while suppressing "bad" uncertainty that leads to instability, rather than collapsing the model into a single mode.

| Metric | EG-GRPO | T2I-R1 |
|---|---|---|
| Quality Mean | 13.86 | 13.85 |
| **Diversity (Vendi Score)** | **2.593** | 2.592 |

Table 4: Comparison of Diversity (Vendi Score) between EG-GRPO (Ours) and T2I-R1 (Baseline) on a subset of samples with similar quality scores. Our method maintains diversity comparable to the baseline.

## 6.5 ANALYSIS OF ENTROPY

Figure 5 compares the entropy distributions of EG-GRPO and T2I-R1 for textual CoT tokens (left) and image tokens (right). The entropy measures the *conditional uncertainty* of token decoding under a given plan or partial image, reflecting generation stability rather than sample-level diversity.

In Figure 5, EG-GRPO reduces both the mean and variance of entropy, with a markedly stronger effect on image tokens. This indicates more confident and stable visual token generation, where high entropy typically corresponds to decoding instability or local artifacts rather than meaningful variation.

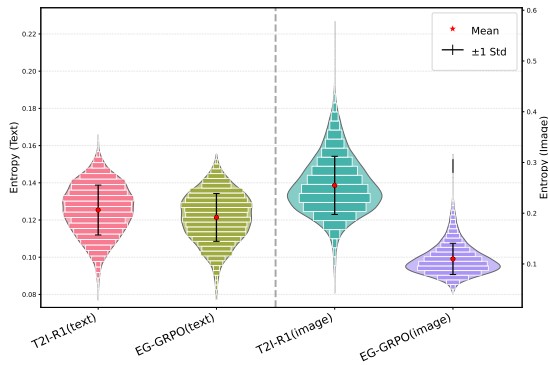

Figure 5: Entropy distributions of EG-GRPO vs. T2I-R1: left for textual CoT tokens, right for image tokens.

Moreover, lower token-level entropy does not imply reduced diversity. EG-GRPO is designed to suppress *bad uncertainty* while preserving *good diversity* across samples, as diversity primarily emerges at the sample level rather than the token level. To avoid premature collapse, we introduce an Entropy Bonus (Eq. 9) that explicitly encourages exploration in high-entropy regions. Consistent with this design, the Vendi Score (as mentioned in Sec. 6.4) shows that sample diversity remains stable throughout training, with only a mild decrease as the model converges to high-reward regions, reflecting a normal exploration–exploitation trade-off rather than mode collapse.

## 6.6 CASE STUDY

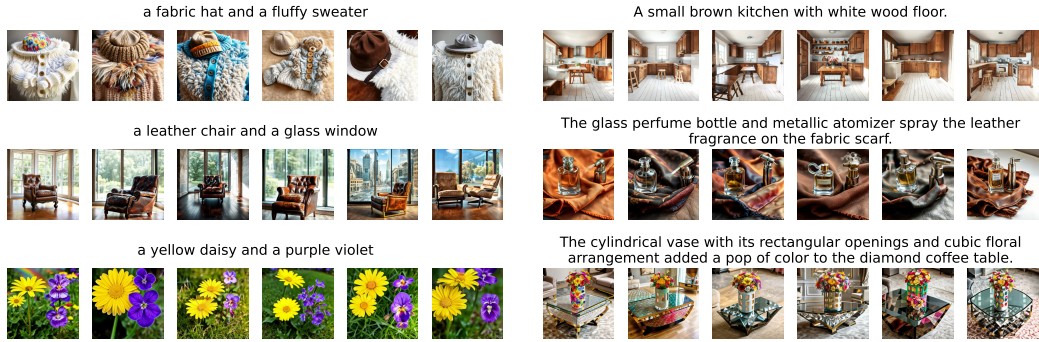

a fabric hat and a fluffy sweater

A small brown kitchen with white wood floor.

a leather chair and a glass window

The glass perfume bottle and metallic atomizer spray the leather fragrance on the fabric scarf.

a yellow daisy and a purple violet

The cylindrical vase with its rectangular openings and cubic floral arrangement added a pop of color to the diamond coffee table.

Figure 6: Qualitative case study of our method on diverse prompts. These results are randomly sampled.

As shown in Figure 6, our method consistently produces high-quality generations across a wide range of prompts. All examples in this case study are randomly sampled. Our method captures fine-grained attributes such as colors and textures with higher fidelity, preserves coherent spatial layouts in complex scenes, and maintains stability when composing multiple objects. These results confirm that entropy-guided optimization enhances both the accuracy and consistency of text-to-image generation.

## 7 CONCLUSION

We studied entropy dynamics in text-to-image generation, showing that CoT expands exploration while reinforcement learning contracts it toward stable, high-reward regions. Both the mean and variance of entropy strongly predict image quality, motivating our Entropy-Guided GRPO (EG-GRPO). By protecting low-entropy tokens and focusing updates on high-entropy ones, EG-GRPO balances stability with structured exploration. Experiments on T2I-CompBench and WISE confirm its state-of-the-art performance and reduced instability, underscoring uncertainty control as a key principle for advancing text-to-image generation.

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

## A  The Use of Large Language Models

In this work, we employed GPT-5 to assist with the polishing of English text. The model was used primarily for improving readability, clarity, and fluency of the written content, ensuring that the final manuscript meets high academic standards.

## B  Proofs and Derivations for EG-GRPO

### B.1  Detailed proof of Proposition 1 (Per-batch budget balance)

**Setup and notation.**  For sequence $i$, write $a^{(i)} \triangleq |A^{(i)}| \geq 0$, $s^{(i)} \triangleq \text{sign}(A^{(i)}) \in \{\pm 1\}$, and $h_t^{(i)} \triangleq \bar{H}_t^{(i)} \in [0, 1]$. With the low/mid/high partitions from Section 5.3, EG-GRPO's per-sequence budget can be written as

$$B_{\text{EG}}^{(i)} = (1 - p_{\text{lo}})\, a^{(i)} + \frac{1}{T^{(i)}} \sum_{t \in \mathcal{S}_{\text{hi}}^{(i,m)}} \Big( \big| s^{(i)} a^{(i)} + \lambda h_t^{(i)} \big| - a^{(i)} \Big). \tag{11}$$

**Lemma 1 (Symmetric sign averaging).**  For any $a \geq 0$ and $b \geq 0$,

$$\frac{1}{2}\Big( |a + b| + |-a + b| \Big) = \max(a, b).$$

*Proof.* If $b \geq a$, then $|a + b| = a + b$ and $|-a + b| = b - a$, so the average is $b$. If $b < a$, then $|a + b| = a + b$ and $|-a + b| = a - b$, so the average is $a$.  □

**Exact decomposition.**  Taking expectation over $s^{(i)}$ (assumed approximately symmetric due to group-normalization), Lemma 1 yields

$$\mathbb{E}_{s^{(i)}}\big[ |s^{(i)} a^{(i)} + \lambda h_t^{(i)}| \big] = \max\big( a^{(i)}, \lambda h_t^{(i)} \big). \tag{12}$$

Plugging into equation 11 and using $(x - a)_+ \triangleq \max(0, x - a)$,

$$\mathbb{E}_{s^{(i)}}\big[ B_{\text{EG}}^{(i)} \big] = (1 - p_{\text{lo}})\, a^{(i)} + \underbrace{\frac{1}{T^{(i)}} \sum_{t \in \mathcal{S}_{\text{hi}}^{(i,m)}} \big( \lambda h_t^{(i)} - a^{(i)} \big)_+}_{\delta^{(i)}(\lambda)}. \tag{13}$$

Averaging over the batch $\mathcal{B}$ gives

$$\mathbb{E}_{\mathcal{B}}\big[ B_{\text{EG}}^{(i)} \big] = (1 - p_{\text{lo}})\, \mathbb{E}_{\mathcal{B}}[a^{(i)}] + \mathbb{E}_{\mathcal{B}}\big[ \delta^{(i)}(\lambda) \big]. \tag{14}$$

Here $\delta^{(i)}(\lambda)$ is nondecreasing in $\lambda$ and equals zero at $\lambda = 0$.

**Target equation for exact $\kappa$-scaling.**  If one desires $\mathbb{E}_{\mathcal{B}}[B_{\text{EG}}^{(i)}] = \kappa\, \mathbb{E}_{\mathcal{B}}[a^{(i)}]$, then equation 14 is equivalent to

$$\sum_{i \in \mathcal{B}} \frac{1}{T^{(i)}} \sum_{t \in \mathcal{S}_{\text{hi}}^{(i,m)}} \big( \lambda h_t^{(i)} - a^{(i)} \big)_+ = (\kappa - 1 + p_{\text{lo}}) \sum_{i \in \mathcal{B}} a^{(i)}. \tag{15}$$

The left-hand side is continuous, nondecreasing in $\lambda$, hence admits a (numerically) unique solution for any $\kappa \in (0, 1]$.

**Closed-form calibration (upper-bound match).**  For an implementable closed form, use $(x - a)_+ \leq x$ with $x = \lambda h_t^{(i)}$:

$$\delta^{(i)}(\lambda) \leq \lambda \cdot \underbrace{\frac{1}{T^{(i)}} \sum_{t \in \mathcal{S}_{\text{hi}}^{(i,m)}} h_t^{(i)}}_{H_{\text{hi}}^{(i)}}.$$

Substituting this into equation 14 yields the upper bound

$$\mathbb{E}_{\mathcal{B}}[B_{\mathrm{EG}}^{(i)}] \le (1 - p_{\mathrm{lo}})\,\mathbb{E}_{\mathcal{B}}[a^{(i)}] + \lambda\,\mathbb{E}_{\mathcal{B}}[H_{\mathrm{hi}}^{(i)}]. \tag{16}$$

Matching the *upper bound* to $\kappa\,\mathbb{E}_{\mathcal{B}}[a^{(i)}]$ leads to the batch-calibrated choice

$$\lambda^{\star} \;\triangleq\; \kappa \cdot \frac{\sum_{i \in \mathcal{B}} a^{(i)} \cdot \frac{1}{T^{(i)}} \sum_{t \in \mathcal{S}_{\mathrm{lo}}^{(i,m)}} 1}{\sum_{i \in \mathcal{B}} \frac{1}{T^{(i)}} \sum_{t \in \mathcal{S}_{\mathrm{hi}}^{(i,m)}} h_t^{(i)}} \;=\; \kappa \cdot \frac{\sum_{i \in \mathcal{B}} |A^{(i)}| \cdot \frac{|\mathcal{S}_{\mathrm{lo}}^{(i,m)}|}{T^{(i)}}}{\sum_{i \in \mathcal{B}} \frac{1}{T^{(i)}} \sum_{t \in \mathcal{S}_{\mathrm{hi}}^{(i,m)}} \mathrm{sg}[\bar{H}_t^{(i)}]}, \tag{17}$$

which is exactly equation 8. This calibration equates "saved" low-entropy budget with "reinvested" high-entropy mass in an upper-bound sense; see Remarks below.

**Remarks on calibration accuracy.** (i) Define the nonnegative discrepancy

$$\varepsilon^{(i)}(\lambda) \;\triangleq\; \lambda H_{\mathrm{hi}}^{(i)} - \delta^{(i)}(\lambda) = \frac{1}{T^{(i)}} \sum_{t \in \mathcal{S}_{\mathrm{hi}}^{(i,m)}} \min(\lambda h_t^{(i)},\, a^{(i)}).$$

It vanishes as $\lambda h_t^{(i)} \gg a^{(i)}$ for most high-entropy tokens, and is statistically damped by batch averaging. (ii) For *exact $\kappa$-scaling*, one may solve equation 15 via a 1D root finder; we keep equation 8 for simplicity and stability. (iii) With $\kappa = 1$, equation 8 delivers batch-level budget neutrality in the calibrated upper-bound sense; empirically it closely tracks neutrality while avoiding per-step root solving.

**Conclusion.** Combining equation 14–equation 16 with the calibration above yields $\mathbb{E}_{\mathcal{B}}[B_{\mathrm{EG}}^{(i)}] \approx \kappa\,\mathbb{E}_{\mathcal{B}}[B_{\mathrm{GRPO}}^{(i)}]$, as stated in Proposition 1.

## B.2 PROOF OF COROLLARY 5.1 (FIXED-POINT NEUTRALITY)

Suppose $A^{(i)} \equiv 0$ for all $i$ in a batch. Then $a^{(i)} = 0$, and the numerator of equation 8 vanishes, yielding $\lambda^{\star} = 0$ for any $\kappa \in (0, 1]$. By equation 9, $\tilde{A}_t^{(i)} \equiv 0$ for all tokens $t$, so the loss equation 10 reduces to the reference-policy KL term. Hence any GRPO stationary point remains stationary under EG-GRPO, proving the corollary.

## B.3 OPTIONAL EXACT PER-BATCH SCALING (IMPLEMENTATION NOTE)

If precise $\kappa$-scaling is required, solve the monotone equation equation 15 for $\lambda$ by bisection or Newton's method. This guarantees $\mathbb{E}_{\mathcal{B}}[B_{\mathrm{EG}}^{(i)}] = \kappa\,\mathbb{E}_{\mathcal{B}}[B_{\mathrm{GRPO}}^{(i)}]$ *exactly* per batch, at the cost of a 1D search.

## C  INSTRUCTION FOR WISE

**WISE Instruction**

You are asked to write a concise text description to guide the generation of an image based on this prompt: "{}". Provide a brief, precise visualization of all elements in the prompt. Your description should:

1. Include every object mentioned.
2. Specify visual attributes (color, number, shape, texture) if given.
3. Clarify spatial or relational positioning if specified.
4. Be concise ($\leq 50$ words) but include the most common features and states inferred from real-world knowledge.
5. Apply real-world knowledge (cultural, religious, temporal, spatial, biological, physical, or chemical reasoning) and select only the single most relevant aspect that naturally enhances the original prompt to infer context (e.g., season, appearance, identity, cultural usage, or natural state) and reflect it in the objects. Use direct, widely accepted interpretations; include cultural or religious meanings only when they are common real-world associations. Avoid abstract or purely metaphorical interpretations.
6. Emphasize the current state of each object individually, as inferred from its environment or context. Reason separately for each object, considering temporal, cultural, or physical factors, and prioritize states logically implied by the prompt.
7. If multiple objects are present, reason from each object's inherent physical or chemical properties and their interactions with the environment and with all other objects. Ensure that the inferred state, behavior, and interaction of every single object is logically correct and consistent with real-world rules, and clearly describe all differences and interactions relative to each other.
8. Ensure realism and aesthetic quality: all objects and interactions must follow real-world rules and appear visually consistent and appealing.
9. Do not omit objects explicitly mentioned, or add ones not specified or logically inferred.
10. Always preserve and emphasize the original objects and scene as the primary focus.
11. Always output a complete natural language description, never an image or symbolic shorthand.

## D  COMPUTATIONAL EFFICIENCY AND CONVERGENCE ANALYSIS

To address concerns regarding the computational overhead of the proposed EG-GRPO, specifically the token-level entropy computation, percentile thresholding, and batch-level bonus recalibration, we conducted a rigorous benchmarking comparison of our method against the baseline T2I-R1.

### D.1  WALL-CLOCK TIME AND MEMORY OVERHEAD

We measured the training step time and peak GPU memory usage on NVIDIA A100 GPUs under identical experimental settings. As shown in Table 5, EG-GRPO introduces negligible overhead.

| Metric | T2I-R1 (Baseline) | EG-GRPO (Ours) | Overhead |
|---|---|---|---|
| Step Time (s) | 50.20 | 50.88 | +1.35% |
| GPU Memory (GB) | 30.79 | 31.17 | +0.38 GB |

Table 5: Computational overhead comparison between T2I-R1 and EG-GRPO. The overhead introduced by our entropy-guided mechanism is marginal in both time and memory.

The minimal increase in wall-clock time ($\sim 1.35\%$) suggests that the complexity of entropy operations ($O(L \cdot V)$ for sequence length $L$ and vocabulary size $V$) is trivial compared to the computational cost of the model's forward and backward passes. Importantly, this overhead ratio is expected to de-

crease further as model scale increases (e.g., larger hidden dimensions), as the entropy calculation depends only on the output logits and not the model depth or parameter count.

## D.2 CONVERGENCE EFFICIENCY

While the per-step cost is marginally higher, EG-GRPO exhibits significantly better sample efficiency. We compared the reward progression of both models over the same number of training steps. As detailed in Table 6, EG-GRPO consistently achieves higher rewards earlier in the training process.

| Step | 100 | 200 | 400 | 600 | 800 |
|------|-----|-----|-----|-----|-----|
| Reward (T2I-R1) | 2.1105 | 2.1130 | 2.1208 | 2.1339 | 2.1340 |
| Reward (EG-GRPO) | **2.1411** | **2.1836** | **2.1968** | **2.2075** | **2.2117** |

Table 6: Convergence comparison: Reward scores at different training steps. EG-GRPO achieves higher performance consistently, indicating a superior time-to-convergence ratio.

This superior convergence efficiency outweighs the slight computational overhead, making EG-GRPO a more practical choice for large-scale training.

## E QUALITATIVE DIVERSITY CASES

To qualitatively compare the diversity of T2I-R1 and EG-GRPO, we generate 20 samples for each of three identical prompts and concatenate them for side-by-side visualization, as shown in Figure 7. For a fair comparison, samples from both models are selected under comparable generation quality scores.

The soft, plush texture of the teddy bear was a comforting companion
for the children at bedtime.

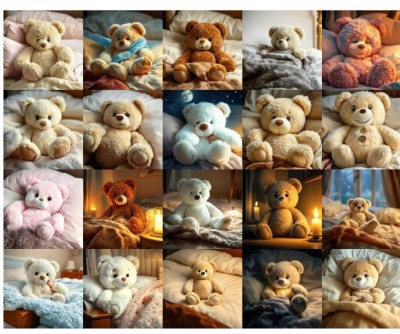 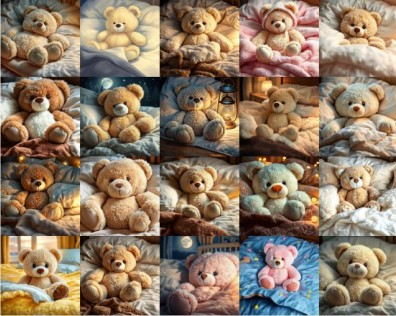

The triangular shelf with its crescent curves and circular brackets held
items in the pentagonal hallway.

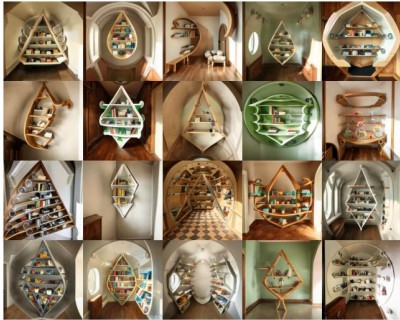 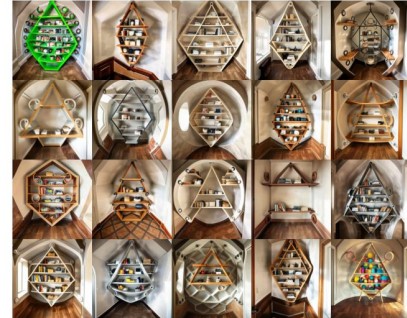

a chicken hidden by a couch

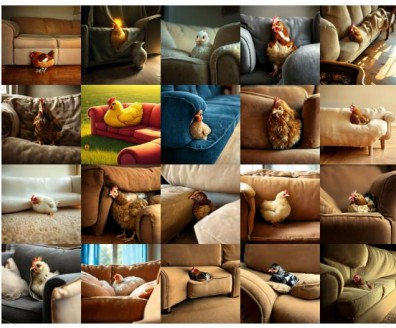 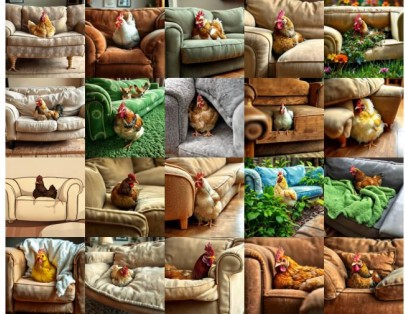

Figure 7: Comparison of generation diversity for **T2I-R1** (left) and **EG-GRPO** (right).

