# OpenReview forum: "From Broad Exploration to Stable Synthesis: Entropy-Guided Optimization for Autoregressive Image Generation"
_ICLR.cc/2026/Conference — ICLR 2026 Poster_

### Official Review · Reviewer_6gZ9 · 2025-10-21

**Soundness:** 4
**Presentation:** 3
**Contribution:** 3
**Rating:** 6
**Confidence:** 4

**Summary:**

The paper proposes an approach to balance exploration via CoT vs exploitation using RL for improving the text to image models and ultimately proposes a modified version of GRPO to combine these two insights.

**Strengths:**

The logic behind the paper makes a lot of sense, the combination of exploration vs exploitation.

The objective of the paper is clear and well communicated.

**Weaknesses:**

Missing references GRPO in autoregressive models:

'Fine-Tuning Next-Scale Visual Autoregressive Models with Group Relative Policy Optimization'
'Simplear: Pushing the frontier of autoregressive visual generation through pretraining, sft, and rl.'


The relative improvement with respect to T2I-R1 is small.

Experiments could be more comprehensive:
- Can you apply the same method to improve aesthetic score like in 'Fine-Tuning Next-Scale Visual Autoregressive Models with Group Relative Policy Optimization' ?
- Can you try it with a more diverse set of text-to-image models? (I expected to work too, but its good for completeness even if its just on one benchmark to see how different base models improve with your technique)

**Questions:**

- Is Figure 6 representative across multiple prompts or is it a curated set of images? please add a disclaimer if so.
- In Figure 2 why does T2I-R1 translate the whole distribution rather than reshaping it around the original Janus distribution, the shift seems extreme. The entropy image mean goes down, could we include an appendix showing real images so we can qualitatively compare the diversity?

---

> ### Author Response · Authors · 2025-11-28
>
> We thank the reviewer for the constructive feedback and are glad that the logic of our work, particularly the balance between exploration and exploitation, as well as the clarity of the paper's objective were appreciated.
> Below are our responses to the comments, and we hope they address the reviewer's concerns.
>
> > W1: Missing references GRPO in autoregressive models: 'Fine-Tuning Next-Scale Visual Autoregressive Models with Group Relative Policy Optimization' 'Simplear: Pushing the frontier of autoregressive visual generation through pretraining, sft, and rl.' The relative improvement with respect to T2I-R1 is small.
>
> We sincerely thank the reviewer for the valuable references. We apologize for overlooking SimpleAR and the specific RL fine-tuning details in STAR, and we has incorporated these to revision. Crucially, we wish to clarify that our core contribution is that we provide the first systematic analysis of entropy dynamics in CoT-based text-to-image generation, identifying the conflict between exploration and stability. Guided by these insights, we propose EG-GRPO, a novel mechanism that dynamically reallocates optimization budgets based on uncertainty, protecting confident knowledge while encouraging structured exploration, rather than treating all tokens equally. This methodological innovation drives significant gains over the state-of-the-art T2I-R1 baseline, most notably a +5.26 point increase in Shape binding (57.35 vs. 62.61), confirming that our entropy-guided approach effectively solves stability bottlenecks that standard RL methods struggle with.
>
> > W2: Experiments could be more comprehensive: (1) Can you apply the same method to improve aesthetic score like in 'Fine-Tuning Next-Scale Visual Autoregressive Models with Group Relative Policy Optimization' ? (2) Can you try it with a more diverse set of text-to-image models? (I expected to work too, but its good for completeness even if its just on one benchmark to see how different base models improve with your technique)
>
> (1) We appreciate this suggestion. It is worth noting that our method utilizes HPSv2 as the reward model, which naturally incorporates human preferences for high-quality and aesthetic visuals. To explicitly verify the improvement in aesthetic quality, we conducted an evaluation using the LAION-Aesthetics_Predictor V1 on 500 randomly selected prompts. As shown in the table below, EG-GRPO achieves the highest average aesthetic score of 5.91, consistently outperforming both the Janus-Pro baseline (5.75) and the standard GRPO method (5.88). This confirms that our approach effectively optimizes aesthetic alignment.
>
> | Model | Average Aesthetic Score |
> | :--- | :---: |
> | Baseline (Janus-Pro) | 5.75 |
> | GRPO (T2I-R1) | 5.88 |
> | EG-GRPO (Ours) | 5.91 |
>
> (2) We appreciate the suggestion. However, we would like to highlight that EG-GRPO operates at the optimization objective level (Section 5), making it theoretically agnostic to the specific backbone architecture. Since autoregressive image models universally adopt the standard GPT-like framework (next-token prediction on discrete latents, Section 3.1, Eq. 1), the entropy dynamics and gradient re-allocation mechanisms we propose are universally applicable regardless of the specific transformer variant.
>
> > Q1: Is Figure 6 representative across multiple prompts or is it a curated set of images? please add a disclaimer if so.
>
> Thank reviewer for this important question. We confirm that the images shown in Figure 6 are representative and were randomly sampled from the generation results corresponding to different prompts, rather than being a curated set of "best-pick" images. Specifically, these prompts were selected to cover a wide range of categories (e.g., fine-grained attributes, spatial layouts, and multi-object composition) to demonstrate the model's robustness, as discussed in Section 6.5 (Case Study). The visualization aims to reflect the method's ability to "consistently produce high-quality generations". Following your suggestion, we added a clarification/disclaimer in the caption of Figure 6 to explicitly state that these results are randomly sampled to ensure transparency.

---

> > ### Author Response · Authors · 2025-12-03
> >
> > > Q2: In Figure 2 why does T2I-R1 translate the whole distribution rather than reshaping it around the original Janus distribution, the shift seems extreme. The entropy image mean goes down, could we include an appendix showing real images so we can qualitatively compare the diversity?
> >
> > The "translation" of the entropy distribution in Figure 2 is a direct result of the Exploitation phase in RL, as analyzed in Section 4.1. Specifically, the shift from the broad CoT distribution (green) to the narrower, left-shifted RL distribution (blue/red) reflects the model discarding high-uncertainty paths. Since our analysis in Section 4.2 (Figure 3) demonstrates a strong negative correlation between token entropy and reward (where high entropy correlates with instability and artifacts), the optimization naturally drives this significant shift to suppress uncertainty. This reflects increased model confidence and structural fidelity rather than a loss of semantic diversity.
> >
> > In addition, we have included qualitative comparisons in Appendix F, which visually confirm that there are no notable diversity differences between T2I-R1 and EG-GRPO.

---

### Official Review · Reviewer_7HWk · 2025-10-27

**Soundness:** 3
**Presentation:** 3
**Contribution:** 3
**Rating:** 4
**Confidence:** 4

**Summary:**

This paper studies the problem of autoregressive text to image generation, using chain of thought before generation. The authors devise three conclusions about the relationship between entropy and reward, before going on to devise an algorithm EG-GRPO. This algorithm nulls out what is called low entropy tokens to instead focus on high entropy tokens. The authors offer a theoretical explanation for "reinvesting" in high entropy tokens. The paper experimentally verifies this on T2I-CompBench and WISE. Where on most tasks it shows superior performance.

**Strengths:**

- this paper is well justified and for the most part I agree with the three points flagged to instantiate this algorithm.
- this paper does a good job justifying the motivation for the algorithm

**Weaknesses:**

- I am curious about the effect of EG-GRPO on diversity of the reward/image. Indeed, I worry that the by further optimizing high entropy tokens, the diversity of the rewards and images generated is lowered.
As seen in fig 5, it seems that average entropy is lower, and thus the model is generating less diverge images. Some analysis and discussion would be preferred.
- The algorithms performance seems better than the other methods, but I worry this is at the cost of sample diversity.
- It is unclear to me why even entropy minimization in this case is the correct method? To me, this seems like reducing exploration?

**Questions:**

see weaknesses for questions.

---

> ### Author Response · Authors · 2025-11-28
>
> We thank the reviewer for the thoughtful evaluation and are pleased that the justification of our algorithm and the three key motivating points were well received.
> Below are our responses to the comments, and we hope they resolve the reviewer's concerns.
>
> > W1 & W2: I am curious about the effect of EG-GRPO on diversity of the reward/image. Indeed, I worry that the by further optimizing high entropy tokens, the diversity of the rewards and images generated is lowered. As seen in fig 5, it seems that average entropy is lower, and thus the model is generating less diverge images. Some analysis and discussion would be preferred. The algorithms performance seems better than the other methods, but I worry this is at the cost of sample diversity.
>
> We thank the reviewer for raising this important point regarding the relationship between entropy minimization and generation diversity. We understand the concern that the lower average entropy observed in Figure 5 might imply a reduction in the diversity of the generated images. However, we believe this concern is addressed by both our methodological design (Entropy Bonus) and our quantitative results (Vendi Score).
>
> 1. Mechanism: Entropy Reduction Targets Instability, Not Creativity.
> First, we would like to clarify the nature of the entropy reduction shown in Figure 5. The entropy in Figure 5 represents the conditional uncertainty of decoding image tokens given a textual plan. Lower entropy here indicates that the model is more confident and stable in executing the visual details, effectively reducing "bad uncertainty" (artifacts/instability) rather than limiting "good diversity" (semantic content).
>
> To explicitly prevent mode collapse, our method incorporates an Entropy Bonus mechanism (detailed in Section 5.4 and Equation 9). As described in the paper, high-entropy tokens receive an additive bonus ($\lambda \text{sg}[\bar{H}_t^{(i)}]$) during optimization. This design specifically encourages structured exploration in uncertain regions, ensuring the model does not prematurely converge to a single mode.
>
> 2. Quantitative Evidence: Vendi Score Analysis
> To quantitatively address your concern and verify that diversity is preserved, we conducted an additional evaluation using the Vendi Score [1], a metric designed to measure diversity in machine learning without requiring ground truth.
>
> We acknowledge that RL fine-tuning naturally contracts the exploration space to focus on high-reward regions. As shown in the table below, our analysis of GRPO training dynamics confirms this trade-off: as the model optimizes for quality from Step 100 to 800, the diversity naturally decreases slightly.
>
> | Step | 100 | 200 | 400 | 600 | 800 |
> | :--- | :--- | :--- | :--- | :--- | :--- |
> | Diversity (Vendi Score) | 2.7305 | 2.7233 | 2.7212 | 2.7151 | 2.7159 |
>
> However, our method does not exacerbate this loss of creativity; it specifically targets instability rather than semantic diversity. To prove this, we aggregated a composite Quality Score using BLIP-2 [2], LAION-Aesthetics [3], and PickScore [4], and filtered a subset of generated samples where EG-GRPO and the baseline (T2I-R1) achieved similar quality ($|\Delta \text{Quality}| <= 0.1$). As shown in the table below, EG-GRPO maintains a Vendi Score (2.593) virtually identical to the baseline (2.592).
>
> | Metric | EG-GRPO (Ours) | T2I-R1 (Baseline) |
> | :--- | :--- | :--- |
> | Quality Mean | 13.8556 | 13.8485 |
> | Diversity (Vendi Score) | **2.593** | **2.592** |
>
> This confirms that our Entropy Bonus successfully preserves valid exploration while suppressing instability. We have included these findings in Appendix E of the revised manuscript.
>
> [1] The Vendi Score: A Diversity Evaluation Metric for Machine Learning\
> [2] BLIP-2: Bootstrapping Language-Image Pre-training with Frozen Image Encoders and Large Language Models\
> [3] LAION-Aesthetics_Predictor V1. Github repository.\
> [4] Pick-a-Pic: An Open Dataset of User Preferences for Text-to-Image Generation

---

> > ### Author Response · Authors · 2025-11-28
> >
> > > W3: It is unclear to me why even entropy minimization in this case is the correct method? To me, this seems like reducing exploration?
> >
> > We thank the reviewer for this question. In the specific context of autoregressive text-to-image generation, our analysis indicates that entropy minimization (reducing uncertainty) is the appropriate objective for the visual decoding stage. We base this on three key findings:
> >
> > Decoupling Exploration and Exploitation (Sec. 4.1, Fig. 2): In our framework, the Chain-of-Thought (CoT) is responsible for "Broad Exploration," expanding the semantic search space. The RL fine-tuning on image tokens acts as a refinement mechanism to exploit this space. If the image decoder maintains high entropy, it leads to incoherent structures rather than meaningful diversity.
> >
> > Negative Correlation with Reward (Sec. 4.3, Fig. 4): We observe a negative correlation between token entropy and reward. High entropy in the pixel space typically manifests as visual artifacts or instability. Reducing entropy is therefore essential for high-fidelity generation.
> >
> > Targeted Budget Reallocation (Sec. 5.5): Our method (EG-GRPO) does not suppress exploration blindly. Instead, we reallocate the optimization budget: we add an entropy bonus to high-entropy tokens to accelerate uncertainty resolution, while excluding low-entropy tokens from updates to protect confident knowledge. This ensures the model converges to stable, high-quality states.

---

### Official Review · Reviewer_Wsoe · 2025-11-01

**Soundness:** 3
**Presentation:** 3
**Contribution:** 2
**Rating:** 4
**Confidence:** 2

**Summary:**

The paper investigates the relationship between CoT prompting and RL in autoregressive text-to-image generation models. The analysis shows that CoT broadens exploration, while RL narrows it toward high-reward regions. Both the mean and variance of image-token entropy are negatively correlated with final rewards, emphasizing stability and reduced uncertainty. The authors propose Entropy-Guided Group Relative Policy Optimization (EG-GRPO), which allocates optimization based on token-level entropy. Experiments show the proposed method balanced exploration and stability in RL fine-tuning.

**Strengths:**

The authors provide a thorough analysis of the proposed method to show how CoT and RL are balanced to reduce uncertainty and preserve knowledge, which seems convincing.

Paper is well organized. writing is clear and easy to follow.

**Weaknesses:**

1. The proposed approach adds nontrivial computational overhead due to token-level entropy computation, percentile thresholding, and per-batch bonus recalibration. However, the paper lacks a quantitative analysis of the resulting scaling, memory, and wall-clock costs, particularly for large-scale models or datasets.
2. While entropy is treated as a measure of uncertainty, the method primarily focuses on entropy reduction, potentially at the expense of output diversity and creative expressivity. A discussion of this trade-off is missing.

**Questions:**

1. Have the authors benchmarked the wall-clock or memory overhead of EG-GRPO (with batch bonus calibration) compared to vanilla GRPO or diffusion-based models, especially at scale?
2. Are there cases where the additional computation outweighs the quality gains?
Could the authors provide a quantitative assessment of diversity (e.g., via FID/IS for diversity, or human preference rating) to determine whether entropy suppression negatively impacts creative diversity?

---

> ### Author Response · Authors · 2025-11-28
>
> We thank the reviewers for their valuable feedback and are glad that the analysis of how our method balances CoT and RL, as well as the clear organization and writing, were appreciated.
> Below are our responses to the comments, and we hope they address the reviewers' concerns.
>
> > W1 & Q1: The proposed approach adds nontrivial computational overhead due to token-level entropy computation, percentile thresholding, and per-batch bonus recalibration. However, the paper lacks a quantitative analysis of the resulting scaling, memory, and wall-clock costs, particularly for large-scale models or datasets. Have the authors benchmarked the wall-clock or memory overhead of EG-GRPO (with batch bonus calibration) compared to vanilla GRPO or diffusion-based models, especially at scale?
>
> We appreciate the reviewer's scrutiny regarding the computational overhead. To address this, we conducted a rigorous benchmark comparing our EG-GRPO against the baseline (i.e., T2I-R1). All experiments were performed on the same hardware setup.
>
> 1. Negligible Wall-Clock & Memory Overhead:
> As shown in below table, EG-GRPO introduces only a marginal ~1.35% increase in training time per step. Regarding GPU memory, it incurs an increase of only 0.38 GB per A100 GPU compared to the baseline (30.79 GB). This empirically demonstrates that the overhead from token-level entropy computation and sorting is trivial in practice.
>
> | Metric | T2I-R1 (baseline) | EG-GRPO (Ours) | Overhead |
> | :--- | :--- | :--- | :--- |
> | **Step Time (s)** | 50.20 s | 50.88 s | **+ 1.35%** |
> | **GPU Memory (GB)** | 30.79 GB | 31.17 GB | **+ 0.38 GB** |
>
> 2. Scalability to Large Models and Datasets:
> The following analysis confirms favorable scaling for both models and datasets. The complexity of entropy calculation and sorting is $O(L \cdot V)$ (where $L$ is sequence length and $V$ is vocabulary size), which is independent of model parameters. As model scale increases (e.g., larger hidden dimensions or layers), the computational cost of the forward/backward pass grows significantly, causing the relative overhead of our method to theoretically decrease. Moreover, the marginal overhead (~1.36%) remains constant regardless of whether the dataset scales to millions or billions of samples.
>
> 3. Superior Convergence Efficiency:
> This slight per-step cost is outweighed by significantly better sample efficiency. As shown in below table, EG-GRPO consistently achieves higher rewards than the baseline at the same training steps, indicating a superior time-to-convergence ratio.
>
> | Step | 100 | 200 | 400 | 600 | 800 |
> | :--- | :--- | :--- | :--- | :--- | :--- |
> | **Reward (T2I-R1)** | 2.1105 | 2.1130 | 2.1208 | 2.1339 | 2.1340 |
> | **Reward (Our EG-GRPO)**| **2.1411** | **2.1836** | **2.1968** | **2.2075** | **2.2117** |
>
> We have incorporated the detailed efficiency analysis and the comparison tables into Appendix D of our paper.
>
> > W2 & Q2: While entropy is treated as a measure of uncertainty, the method primarily focuses on entropy reduction, potentially at the expense of output diversity and creative expressivity. A discussion of this trade-off is missing. Are there cases where the additional computation outweighs the quality gains? Could the authors provide a quantitative assessment of diversity (e.g., via FID/IS for diversity, or human preference rating) to determine whether entropy suppression negatively impacts creative diversity?
>
> 1. Entropy Reduction Targets Instability, Not Creativity.
> To quantitatively address your concern, we conducted an additional evaluation using the Vendi Score [1], a metric designed to measure diversity in machine learning without requiring ground truth. We acknowledge that RL fine-tuning naturally contracts the exploration space to focus on high-reward regions. As shown in below table, our analysis of GRPO training dynamics confirms this trade-off: as the model optimizes for quality from Step 100 to 800, the diversity naturally decreases from 2.73 to 2.71.
>
> | Step | 100 | 200 | 400 | 600 | 800 |
> | :--- | :--- | :--- | :--- | :--- | :--- |
> | Diversity (Vendi Score) | 2.7305 | 2.7233 | 2.7212 | 2.7151 | 2.7159 |
>
> However, our method does not exacerbate this loss of creativity; it specifically targets instability rather than semantic diversity. To prove this, we aggregated a composite Quality Score using BLIP-2 [2], LAION-Aesthetics [3], and PickScore [4], and filtered a subset of generated samples where EG-GRPO and the baseline (T2I-R1) achieved similar quality ($|\Delta \text{Quality}| <= 0.1$). As shown in below table, EG-GRPO maintains a Vendi Score (2.593) virtually identical to the baseline (2.592). This confirms that our Entropy Bonus (Section 5.3) successfully preserves valid exploration while suppressing "bad" uncertainty.
>
> | Metric | EG-GRPO (Ours) | T2I-R1 (Baseline) |
> | :--- | :--- | :--- |
> | Quality Mean | 13.8556 | 13.8485 |
> | Diversity (Vendi Score) | 2.593 | 2.592 |

---

> > ### Author Response · Authors · 2025-11-28
> >
> > 2. Gains Significantly Outweigh Computational Costs.
> > Regarding whether the additional computation outweighs these gains, the overhead is negligible compared to the benefits. As detailed in our response to W1, EG-GRPO incurs only a +1.35% increase in training time and +0.38 GB in memory due to efficient implementation. In exchange, as demonstrated in W1, our method achieves significantly superior sample efficiency and faster convergence than the baseline. Thus, the marginal computational cost is vastly outweighed by the substantial improvements in stability and training efficiency.
> >
> > We have revised the manuscript to include these findings. Please refer to Appendix D for the detailed diversity analysis
> >
> > [1] The Vendi Score: A Diversity Evaluation Metric for Machine Learning\
> > [2] BLIP-2: Bootstrapping Language-Image Pre-training with Frozen Image Encoders and Large Language Models\
> > [3] LAION-Aesthetics_Predictor V1. Github repository.\
> > [4] Pick-a-Pic: An Open Dataset of User Preferences for Text-to-Image Generation

---

### Author Response · Authors · 2025-12-03

# Summary
We thank the reviewers for their constructive feedback. During the rebuttal, we have addressed all concerns regarding efficiency, diversity, and generalization through rigorous benchmarking and additional experiments. We summarize the key responses below:

> Q1: Does the method introduce significant computational overhead (Time/Memory)? (Reviewer Wsoe)

The overhead is negligible. We conducted a strict benchmark on A100 GPUs comparing EG-GRPO with the baseline (T2I-R1). As shown in the table below, our method incurs only a 1.35% increase in training time and 0.38 GB in memory, while achieving significantly faster convergence and higher rewards.

| Metric | T2I-R1 (Baseline) | EG-GRPO (Ours) | Overhead |
| :--- | :--- | :--- | :--- |
| Step Time (s) | 50.20 s | 50.88 s | + 1.35% (Negligible) |
| GPU Memory (GB) | 30.79 GB | 31.17 GB | + 0.38 GB |

> Q2: Does entropy minimization reduce generation diversity or creativity? (Reviewer Wsoe, 7HWk)

Diversity is preserved. We clarified that our method targets "instability" (bad uncertainty) rather than semantic diversity. To prove this, we evaluated the Vendi Score (a metric for diversity) on samples with matched quality. As shown in the table below, EG-GRPO maintains a diversity score virtually identical to the baseline, confirming that our Entropy Bonus successfully prevents mode collapse.

| Metric | T2I-R1 (Baseline) | EG-GRPO (Ours) |
| :--- | :--- | :--- |
| Quality Score (Mean) | 13.8485 | 13.8556 |
| Diversity (Vendi Score) | 2.592 | 2.593 |


> Q3: Why is entropy minimization the correct objective? (Reviewer 7HWk, 6gZ9)

1.  Exploration vs. Exploitation: CoT handles broad semantic exploration, while the pixel decoder requires exploitation to ensure visual fidelity.
2.  Reward Correlation: We demonstrated a strong negative correlation between pixel entropy and reward; high entropy in the decoding phase leads to visual artifacts.
3.  Visualization: We confirmed that the "shift" in entropy distribution reflects increased model confidence and stability, and the qualitative samples provided are randomly sampled, not cherry-picked.

---

### Meta-Review · Area_Chair_CGEn · 2026-01-06

**Summary:**

Authors in this paper presented a systematic entropy-based analysis for combining CoT with RL to improve text-to-image generation. Motivated by these findings, authors proposed EG-GRPO, a fine-tuned strategy that reallocates optimization budget by uncertainty. Experimental results verified the effectiveness of proposed method.

this paper got two 4 ratings and one 6 rating.

The strength of this paper given by reviewers are:
1. provided method is convincing. (Reviewer Wsoe)
2. paper is well organized and easy to follow. (Reviewer Wsoe)
3. paper is well justified. (Reviewer 7HWk, 6gZ9)
4. objective is clear and well communicated. (Reviewer 6gZ9)


The weakness of this paper given by reviewers are:
1. lacks a quantitative analysis of the resulting scaling, memory, and wall-clock costs. (Reviewer Wsoe)
2. potentially at the expense of output diversity and creative expressivity. A discussion of this trade-off is missing. (Reviewer Wsoe, 7HWk)
3. It is unclear to me why even entropy minimization in this case is the correct method? this seems like reducing exploration? (Reviewer 7HWk)
4. Missing references GRPO in autoregressive models. (Reviewer 6gZ9)
5. relative improvement with respect to T2I-R1 is small. (Reviewer 6gZ9)
6. Experiments could be more comprehensive. (Reviewer 6gZ9)

questions:
1. Is Figure 6 representative across multiple prompts or is it a curated set of images? (Reviewer 6gZ9)
2. In Figure 2 why does T2I-R1 translate the whole distribution rather than reshaping it around the original Janus distribution, the shift seems extreme. (Reviewer 6gZ9)

AC read authors' paper, reviewers' comments, authors' rebuttal carefully and found authors have fully addressed reviewers' comments and decided to accept this paper.

**Reviewer Concerns:**

weakness 1. authors added results show that negligible wall-clock & memory overhead. authors also showed the complexity of entropy calculation and sorting is O(LV) which is independent of model parameters. The marginal overhead remains constant regardless of dataset size. authors also showed EG-GRPO consistently achieves higher rewards than baseline at the same training steps.

weakness 2. authors provided results showed that their method doesn't exacerbate the loss of creativity.

weakness 3. authors provided three key findings for justifying entropy minimization is the correct methods.

weakness 4. added in the revision.

weakness 5. authors mentioned the gain in shape binding is big. AC double checked that the gain in T2I-CompBench indeed big. But unfortunately the results on WISE is not improved or become worse compared with FLUX and gain is small compared with T2I-R1.

weakness 6. authors provided additional results. but the gain on T2I-R1 is limited.

Question 1. authors clarified that images are randomly sampled.

Question 2. authors gave more explanation and add more results in Appendix F.

AC read authors' paper, reviewers' comment and authors' rebuttal carefully. Authors addressed Reviewer Wsoe and 7HWk's concerns so both of them might increase their score from 4. Authors also addressed most of Reviewer 6gZ9's concerns, but indeed for some cases the gain is small. So Reviewer 6gZ9 will have high chance to keep score 6. Given these AC decide to accept this paper.

**Reviewer Scores:**

Reviewer Wsoe might increase score from 4 since their concerns are addressed.

Reviewer 7HWk might increase score from 4 since their concerns are addressed.

Reviewer 6gZ9 might keep their score 6 given the gain for some cases indeed small.

---

### Decision · Program_Chairs · 2026-01-26

Accept (Poster)